# Application of Silicon Nanowire Field Effect Transistor (SiNW-FET) Biosensor with High Sensitivity

**DOI:** 10.3390/s23156808

**Published:** 2023-07-30

**Authors:** Huiping Li, Dujuan Li, Huiyi Chen, Xiaojie Yue, Kai Fan, Linxi Dong, Gaofeng Wang

**Affiliations:** 1Ministry of Education Engineering Research Center of Smart Microsensors and Microsystems, School of Electronic Information, Hangzhou Dianzi University, Hangzhou 310018, China; 2The Children’s Hospital of Zhejiang University School of Medicine, Hangzhou 310052, China; 3School of Automation, Hangzhou Dianzi University, Hangzhou 310018, China

**Keywords:** silicon nanowires (SiNWs), field effect transistors (FETs), biosensor, biomedical detection, surface modification

## Abstract

As a new type of one-dimensional semiconductor nanometer material, silicon nanowires (SiNWs) possess good application prospects in the field of biomedical sensing. SiNWs have excellent electronic properties for improving the detection sensitivity of biosensors. The combination of SiNWs and field effect transistors (FETs) formed one special biosensor with high sensitivity and target selectivity in real-time and label-free. Recently, SiNW-FETs have received more attention in fields of biomedical detection. Here, we give a critical review of the progress of SiNW-FETs, in particular, about the reversible surface modification methods. Moreover, we summarized the applications of SiNW-FETs in DNA, protein, and microbial detection. We also discuss the related working principle and technical approaches. Our review provides an extensive discussion for studying the challenges in the future development of SiNW-FETs.

## 1. Introduction

A biosensor is an integrated receptor–transducer device capable of transforming a biological reaction into a measurable signal. As an analytical device, biosensors are widely used for biological sample analysis. Traditional biosensors have two components: the receptor and the sensor detection device. In general, receptors are small molecular compounds such as antibodies, enzymes, or DNA molecules, which can recognize antigen, protein, microbial and DNA, etc. Meanwhile, trace amounts of current, voltage, optical, or impedance signals [1] are read out by sensor detection devices. Up to now, the detection performance of chemical sensors or biosensors has been greatly improved due to the appliance of nanotechnology. Nanotechnology is extensively used to study the properties and applications of materials ranging from 1 to 100 nanometers [2]. In the nanometer scale, the physical and chemical properties of materials are mainly affected by a high surface-to-volume ratio, which may shorten the time of detection [3].

In recent years, the nanomaterials, including nanoparticles, nanowires (NWs) [4,5], and carbon nanotubes (CNTs) [6], have provided a new approach for biosensing due to their size in comparison to analytical targets. They have been widely used in chemical analysis [7], medical diagnosis [8], and food safety testing [9]. For example, multiwalled carbon nanotubes (MWCNTs), SnO_2_ nanoparticles, and chitosan can be used for the detection of pesticide residue [10]. In addition, nanoparticles can serve as an alternative to fluorescent dyes. However, it is time-consuming and probably causes conformational changes or steric hindrance. Because nanoparticles need to go through a series of labeling processes when used in fluorescent dyes, the spatial conformational change occurs when the surface is fixed to adapt, resulting in labeled adaptations that are close to the nanoparticles and induce fluorescence resonance energy transfer (FRET) [11]. This increases the operational complexity and time consumption of the process of building biosensors. Meanwhile, it is very difficult for the precise preparation of CNTs due to the unskilled interface protocol for extensive analyte binding [12].

In addition to performance factors such as sensitivity and specificity, economic cost is another key factor that must be considered in the development and fabrication of biosensors. Nanowires are becoming an excellent choice for biomedical sensor fabrication due to their sensitivity to biological targets and manufacturability in large-scale production. Therefore, the SiNW-FET biosensors, which combined nanotechnology with semiconductor silicon processing methods, have caused wide concern since they were first reported in 2001 [5].

Classically, for the fabrication of SiNW-FETs, there are two main techniques known as “bottom-up” and “top-down”. Also, the SiNWs (light doping) and source/drain electrodes (heavy doping) were defined. Like most electrochemical biosensors [13], SiNW-FETs are composed of three electrode structures: the source electrode, drain electrode, and gate electrode (back gate or solution gate). As the sensing element, SiNW is modified with small biological molecules by a series of modification methods to match the target object. The conductance modulation of SiNW-FETs can be understood by the change in charge density. Furthermore, capturing targets on the sensor surface could induce a change in the internal charge density of the devices. 

At present, SiNW-FETs have emerged as an excellent technology in the field of biological research, with their superior physical properties such as high carrier mobility [14], high current switch ratio [15], and close to ideal subthreshold slope [16]. Furthermore, the materials are available at a low cost. They have been successfully applied in the detection and recognition of special proteins [17], DNA [18,19], or RNA as cancer or cell markers [2,20]. Numerous studies have reported the great achievement in the manufacturing [21] and surface modification [22] of SiNW-FETs, and the sensitivity of which has been confirmed to reach the attomole level [23]. Thus, SiNW-FETs possess wide application prospects in biosensing and biomedical fields. 

Here, we summarize the recent progress of the applications of SiNW-FETs on the detection of DNA sequences, proteins, and microorganisms, including bacteria and viruses. The repetitive surface modification and stability of SiNW-FETs are also discussed. We introduce the working principle and surface modification technique of SiNW-FETs, followed by the challenges of surface modification technology and high-throughput multiplex detection. 

## 2. Working Principles of SiNW-FETs

As typical FET-based devices, SiNW-FET biosensors are composed of three electrode structures: the source electrode, drain electrode, and gate electrode. The source and drain electrodes are both linked with semiconductor channels, which are made of SiNW. At the same time, the source–drain current is adjusted over the channel conductance via a varying bias voltage on the gate electrode. As an effective sensing element, SiNW can sensitively sense the change in the external electric field owing to its properties of field emission and electron transport. 

The typical structure and sensing principle of SiNW-FET biosensors for identifying biomolecules are shown in Figure 1. Firstly, the target is captured by the receptor and anchored on the SiNW surface, and the data acquisition (DAQ) system supplies gate voltage (Vg) to ensure a stable output signal. Due to the charged nature of biomolecules such as DNA, the binding of the charged target to a receptor can alter the gate potential (or conductivity). The resulting signals [24] are electrically measured by a lock-in amplifier, while the sensing part of the device is usually immersed in a phosphate-buffered saline solution (PBS) to maintain the biological activity of detecting targets. For example, Cao et al. [25] used a 1 μM ssDNA-PBS solution during the SiNW-FET surface modification to maintain the biological activity of ssDNA, thereby ensuring a successful detection. 

To efficiently capture the targets, bioreceptors that are capable of specifically recognizing targets are modified onto the SiNW surface. The target–receptor interaction induces the internal charge (electron or hole) distribution (accumulation or depletion) of SiNW-FET devices, thereby affecting the source–drain current. In fact, as shown in Figure 1, the direction of conductance during this change reflects the polarity (positive or negative) of the charge carried by the target, and the change in conductance reflects the target–receptor interaction.

The SiNW-FET biosensor is often used for immunoassays and is classified into an n-type or p-type according to its structural properties. For the n-type SiNW-FET device, when positively charged target molecules combine with the receptors modified on the SiNW surface, the positive charges in target molecules induce negative charges in SiNW. The electrons will accumulate in the channel, further enhancing the conductance or the source–drain current. Conversely, when the target molecule is negative, the reduction in electrons leads to a decrease in the conductivity and current. This has been demonstrated by Gao et al. [26]. They found that in the subthreshold state of the n-type SiNW-FET biosensor, the higher the negative charge concentration, the more obvious the change in the relative current. A significant increase in resistance was observed immediately upon the introduction of the DNA probe onto SiNW surface.

For p-type SiNW-FET devices, negatively charged target molecules will increase the conductivity or current since the charge carriers are mainly holes. Conversely, if the target molecules are positive, the holes’ depletion will lead to the depression of the conductance and current. Yang et al. [20] confirmed that the accumulation of the holes in the p-type channel changes the source–drain current, and the amount of negative charges ramp up at a higher pH due to deprotonation. In addition, in order to achieve a lower signal-to-noise ratio, SiNWs are integrated to form a new biosensing device that links to a semiconductor parameter analyzer or another measurement device for real-time monitoring [27,28].

## 3. Surface Modification Methods of SiNW

Surface modification is a critical step in the integration of biorecognition elements and transducers in biosensor fabrication. In SiNW-FET biosensors, SiNWs must be chemically modified to accomplish the integration of bioreceptors and FET sensors. The modification of biorecognition elements (i.e., receptors) onto SiNWs can provide specific binding sites for target molecules. In bioelectrical measurement, target recognition is achieved by SiNWs modified with specific receptors (antibodies, DNA, etc.). Usually, the Si/SiO_2_ NW surface is silanized before receptor modification. Alkoxysilane derivatives can react with hydroxyl groups (OH) on the surface of SiNWs and act as linkers to immobilize receptors onto SiNWs. 3-(trimethoxysilyl) propyl aldehyde (APTMS) is coupled with SiNW surfaces to present a terminal aldehyde group that can directly immobilize DNA [19] and antibodies [28]. In additional, 3-aminopropyltrimethoxysilane (APTES), the most popular linker, consists of a silane group and an amino group. Its amino groups can anchor biotin [29], calmodulin [30,31], and oligonucleotide [32,33,34,35]. Chu et al. [36] soaked the chip in an APTES solution (2% in acetone) at room temperature for one hour to form a single layer, and then modified the DNA probe. In other works in the literature, 3-mercapto propyl trimethoxysilane (MPTMS) has been proposed to form a thiol (SH)-termination to modify the surface. Chen et al. [37] were able to modify DNA probes using a disulfide bond formed on the SiNW surface by MPTMS. 3-glycidyloxy propyltrimethoxysilane (GPTES) is a kind of epoxy silane that can form a homogeneous silane monolayer in buffer solutions (pH 8–9). Schwartz et al. [38] silanized the SiNW device with GPTES directly. Amino-modified DNA sequences can then be directly and covalently attached to the open epoxy ring without any other cross-linking molecules. Figure 2 shows the above silicon nanowire modification method.

The natural oxide layer on the SiNW surface can prevent silanization or even shield the electric fields generated by the analyte, thereby limiting the performance of the sensor. Bunimovich et al. [39] achieved the quantitative and real-time detection of single-stranded oligonucleotides with SiNW in a physiologically relevant electrolyte solution. Compared with SiNW-FETs with an oxide layer, they found that SiNWs without a natural oxide layer exhibited better solution-gated FET characteristics and improved the sensitivity of single-strand DNA detection significantly. Therefore, prior to the silanization of SiNWs, it is necessary to remove the oxide coating by immersing them in a 2% dilute hydrogen fluoride (HF) solution for 3 s. 

Furthermore, treating the SiNW surface with ultraviolet radiation (UV)/O_2_ plasma prior to silanization will remove the surficial inherent impurities and produce rich terminal hydroxyl groups (-OH) on the SiNW surface, thereby improving the hydrophilicity and then improving the efficiency of silanization and the sensing performance. The experimental results of Rahmnan et al. [40] showed that the detection limit of the SiNW-FET sensor without O_2_ plasma treatment was 4.131 × 10^−13^ M, while it could be as low as 1.985 × 10^−14^ M after 60 s of O_2_ plasma treatment. It has been demonstrated that O_2_ plasma treatment of SiNWs prior to silylation can improve the sensitivity of the sensor.

## 4. Application of SiNW-FET Biosensors

### 4.1. DNA Determination

Deoxyribonucleic acid (DNA), one of the most important genetic materials, functions not only as carriers for transferring genetic information, but also as ideal biomarkers for clinical diagnosis. As a good candidate for monitoring DNA/RNA hybridizations with ultra-high sensitivity (femtomolar level), SiNW-FET has higher integration and label-free detection abilities than surface plasmon resonance (SPR) [41] and quartz crystal microbalance [42]. 

Many disease biomarkers are DNA molecules like hepatitis B virus, avian influenza virus, cancer, and others. The detection of specific DNA is extremely important from the perspective of medical diagnoses. For example, the early diagnosis of cancer is crucial to a patient’s life. Li et al. [43] achieved the detection of tumor marker circulating tumor DNA (ctDNA) with a sensitivity of 10 aM using an inverted triangle SiNW array FET (Figure 3a). The cross-sectional dimensions of SiNWs were approximately 90 nm with a deviation less than ±20 nm. Compared with a single SiNW, 120 SiNWs with a high consistency of dimensions enabled the superposition of signals, resulting in the much stronger response signal and a higher signal-to-noise ratio.

Gao et al. [25] detected the DNA of two pathogenic strains of avian influenza (H1N1 and H5N1) by immobilizing two DNA probes on the surface of SiNW, respectively. Their experimental results showed that the current was related to the logarithm of the DNA concentration. Lin et al. [44] used a Poly-SiNW FET to achieve the specific and ultra-sensitive (fM level) detection of pathogenic avian influenza virus (H5 and H7) DNA. 

Due to the large amount of negative charges in the phosphate backbone of DNA/RNA, the charge carrier density of SiNW-FET biosensors varies significantly during DNA or RNA hybridization [19,45]. Although peptide nucleic acid (PNA) has no phosphate groups in its backbone, it was proven to be capable of determining the distance between target–receptor binding sites and the SiNW surface. As shown in Figure 3b, the binding of PNA/DNA or PNA/RNA strands is stronger than DNA/DNA or DNA/RNA. Hahm et al. [19] demonstrated that PNA-DNA duplex formation has successfully contributed to the detection of DNA in real time. They observed a considerable increase in conductance in the p-type PNA-modified SiNW-FET, with the detection limit down to 10 fM. PNA-based SiNW-FETs are a promising detection platform for various physiological applications [34]. Zhang et al. [32] achieved a detection limit down to 1 fM for miRNA sensing in PNA/SiNW-FET (Figure 3b). It is able to identify miRNAs from the whole RNA extract of HeLa cells, which will contribute to the early diagnosis of tumors. DNA extraction is not necessary for testing ctDNA in blood, while it is required before some viral or cellular nucleic acid assays.

Wenga, G. et al. [46] proposed a novel stepped polycrystalline silicon nanowire field effect transistor for the ultrasensitive detection of DNA hybridization. DNA probes were immobilized on the surface of polycrystalline silicon nanowires that were synthesized by sidewall spacer technology to detect DNA targets. The experimental results showed that the detection limit of complementary DNA targets could be 1 fM.

Due to the unique advantages of DNA analogues as probes for detecting target DNA, more and more scholars have begun to pay attention to DNA analogs. Hu, W.P., and colleagues [47] found that the configured neutralizing of DNA analogues as probes for target DNA can increase the Debye length, which is beneficial for improving the sensitivity and signal-to-noise ratio of the biosensor. They synthesized two partially neutralized chimeric DNA products and a fully neutralized DNA sequence using phosphomethylated nucleotides. The design of the neutralizing DNA probe reduced the ion solubility required for hybridization with the target DNA and increased the Debye length. In addition, the uncharged DNA probes avoided charge repulsion. A BTP buffer was also used to reduce the Debye shielding effect. Therefore, the detection limit of the SiNW-FET was as low as 0.1 fM. 

In order to increase the sensitivity and repeatability of the SiNW-FET sensor, Mikael et al. [48] demonstrated repeatable and efficient DNA hybridization detection by coating the SiNW’s surface with a thin layer of HfO_2._ In a dry environment, the SiNWs were passivated by HfO_2_, and the DNA probes were transplanted onto the surface. Because HfO_2_ had a higher density of OH groups and a high dielectric constant, the density of DNA hybridization detected by this sensor was 10^10^ cm^−2^.

Now, SiNW-FET biosensors have been regarded as promising tools in DNA detection due to their label-free, ultra-highly sensitive, and rapid detection capabilities. As discussed above, SiNW-FET has been applied to detect DNA hybridization, infectious viruses, and miRNAs. However, further optimization of the device is still needed, such as reducing the width and length of the sensor or reducing unnecessary procedures to improve the sensitivity and accuracy of nanoscale devices.

Xie et al. [49] constructed an independent and integrated microfluidic nanodetection system that incorporates SiNW-FET biosensors for biodetection and analysis. All analytical processes, such as liquid sample delivery, thermostatic control, signal amplification, data acquisition, and result display, are performed automatically. This portable nanosensor detection system is mainly composed of five parts: the liquid circuit module, the light modulation module, the constant temperature control module, the signal acquisition and amplification module, and the status and result display module. The system can effectively solve the problem that SiNW-FET biosensors usually require discrete detectors and have not yet been implemented in the integrated and mobile fields. Because the surfaces of SiNW-FET biosensors are inevitably exposed to the external environment, they are susceptible to interference. They are extremely sensitive to external environmental factors such as temperature, light, and pH, which can cause detection errors. Therefore, SiNW-FET is expected to detect biomarkers in field applications in the near future.

### 4.2. Protein Detection

Protein is an important component of human cells and tissues, which plays a significant role in human life activities. Proteins realize the expression of genes under the control of DNA, which make organisms exhibit a variety of genetic characteristics. Many biomarkers used in disease diagnosis are proteins [17,50]. Therefore, achieving effective and sensitive detection of proteins as biomarkers is of great significance for the disease diagnosis.

The Enzyme-Linked Immunosorbent Assay (ELISA) [51], laser, and fluorescence technology [52] have all been used for the detection of proteins. Protein detection and separation technology have been constantly evolving due to improvements in microelectronic technology. Nowadays, more and more researchers are focusing on SiNW-FETs, which are suitable for protein detection. 

The real-time detection of biotin using an SiNW-FET biosensor was first achieved in 2001 by Cui et al. [5]. And, the detection limit reached 10 pmol/L, which was far lower than others test method. However, it was not possible to continuously detect samples with different concentrations due to the irreversible combination of biotin and streptomycin. Kim et al. [53] proposed the combination of a conventional immunoassay (Sandwich ELISA) with high-precision FET devices and accomplished the multichannel detection of three different cancer markers (carcinoembryonic antigen (CEA), prostate specific antigen (PSA), and alpha fetoprotein (AFP)) in serum without pretreatment successfully. As illustrated in Figure 4, four different antibodies were self-assembled on the chip surface (three tumor markers PSA, CEA, AFP, and monoclonal antibodies (IgG)). Then, serum antigens at concentrations of 1 and 10 ng/mL were injected into groups with tumor markers fixed, and changes before and after the injection were observed. The detection limit was as low as ng/mL level.

Recently, many emerging SiNW-based biosensors have been designed for in-depth studying in protein detection. Gong et al. [54] performed the ultra-sensitive detection of IgG using the SiNW-FET biosensor. They fabricated a single SiNW-FET, a double SiNW-FET, and a quad SiNW-FET into a single chip and detected the Norovirus DNA and IgG at concentrations of 1 fM and 10 fM, respectively. Then, as shown in Figure 5, Li et al. [55] developed a multichannel dual-gate SiNW-FET chip for dual-channel detection of BC tumor markers. By modifying monoclonal CA15-3 and CEA antibodies on different SiNW surfaces, the SiNW-FET could be used for the dual-channel specific detection of breast cancer tumor markers, CA15-3 and CEA, respectively. The experimental results showed that the double-gated SiNW-FET can amplify the detection signal exponentially through the capacitive coupling effect, thus improving the signal-to-noise ratio of the SiNW-FET. The sensor can accurately detect CA15-3 down to 0.1 U/mL and CEA at 0.01 ng/mL. Hence, compared with the traditional SiNW-FET, the multichannel design method increased the source–drain current signals, effectively restrained the fluctuation noise inside the transistor, and made the detection system more stable.

However, there are performance differences between the SiNW-FET devices manufactured by the same process. Lu et al. developed a novel SiNW biosensor for the detection of human tear MMP-9 based on an optimized fabrication of an optical calibration and low salt concentration scheme, achieving high sensitivity (86.96%) and specificity (90%) in the diagnosis of DED. Their device was manufactured to demonstrate minimal sensor-to-sensor variation, with optical calibration and a low salt concentration protocol allowing consistent response between sensors and sensitive and accurate detection of MMP-9 in human tears with high agreement with ELISA results [56].

SiNW-FET, as a novel kind of biosensor, could be affected by the Debye–Hückel screening effect induced by ions during protein detection [29]. The higher concentration of ions in a protein solution, the shorter the Debye length and the higher the screening efficiency. In other word, the Debye length is inversely proportional to the square root of the ionic strength. In general, because of the electrolytic medium, the interacting electric field exerted from the captured target charges is shielded. Therefore, it is necessary to reduce the Debye–Hückel screening effect when using an SiNW-FET biosensor for protein detection.

The common way to reduce the concentration of ions is by diluting the buffer solution to a low concentration. However, it is still far from effective protein detection. Kim et al. [50] explored the effect of the Debye length on the highly sensitive and label-free detection of cTnI, a biomarker for the diagnosis of acute myocardial infarction. The Debye lengths were calculated for PBS with different ion concentrations (1 × PBS, 0.1 × PBS, 0.01 × PBS3; ionic strength 180, 18, and 1.8 mM, respectively). It was found that the sensitivity increased with the decrease in buffer ion concentration, and the optimal condition was 0.01 × PBS. Therefore, the buffer concentration has a great effect on sensor determination [57,58]. But, extremely low salt concentrations may reduce the activity of proteins, aggravating the difficulty of effective detection.

Chang, S. M et al. [59] reported that the developed SiNW-FET exhibited a lower detection limit of 0.016 ng/mL than previously developed FET devices and showed great potential for future applications in point-of-care (POC) diagnostics. Puppo et al. [60] studied a terminal amnestic-modified SiNW-FET sensor that successfully detected an anti-rabbit antigen in a PBS buffer, further demonstrating the detection of anti-rabbit antigens in a tumor extract solution in the presence of 100,000 non-specific proteins [61,62]. At the same time, Meir developed a dissociative state sensing method that uses temporal differentiation between low-affinity matrix components and high-affinity target molecules as they are dissociated, enabling direct sensing when the biological samples on modifications are rinsed by low ionic strength buffer solutions [63].

It is found that the voltage power spectral density generated by current-biased SiNW-FET is related to 1/f-dependence in the frequency domain. Zheng et al. [64] performed frequency domain detection of a prostate-specific antigen, whose detection sensitivity was 10 times higher than that of time domain detection. This frequency domain measurement method showed the prospect of simple and rapid biomarker screening in medical applications, i.e., in disease diagnosis [60] and drug discovery.

So far, most of the traditional methods for detecting mycobacterium tuberculosis do not meet the requirements of actual TB detection, and the SiNW-FET sensor can solve the appeal problem. Ma [65] developed a fast and sensitive SiNW-FET biosensor for the detection of the Mycobacterium tuberculosis Ag85B protein. The detection limit of MTB Ag85B is as low as 0.01 fg/mL (0.33aM), it has good specificity and stability, and the detection response time is within 30 s. And the detection range is from 1 fg/mL to 100 fg/mL. Compared to other methods, the SiNW-FET biosensor requires only a small sample (7 μL) and can detect the Ag85B protein with a significantly wider dynamic linear range in a shorter time period.

Although the SiNW-FET biosensors were mostly used in laboratory work, they demonstrated high detection sensitivity, specificity, and relativity, which have potential for practical applications such as food safety detection and clinical diagnosis.

### 4.3. Microbiological Detection

A microorganism is an organism that can be seen only through a microscope, and they are pervasive in life. In fact, any environment that is devoid of microorganisms is certainly the exception rather than the rule. Microorganisms include bacteria, viruses, and fungi. A pathogenic organism is an organism that is capable of causing diseases in a host (person). So far, there have been many large-scale infection outbreaks caused by pathogenic microorganisms. Examples include swine flu in Mexico in 2009, Ebola in West Africa in 2014, and the plague in Madagascar in 2017. And especially the COVID-19 outbreak, which started in 2020, has caused global infections. The large-scale infection outbreak of these pathogenic microorganisms seriously threatens people’s health and life. So, highly sensitive detection methods are essential for the detection of bacteria and viruses to prevent the spread of infection.

Molecular detection methods are used for the detection of microorganisms. A polymerase chain reaction (PCR) [34] was used in an amplifier to amplify specific DNA fragments, and could be used for bacteria or virus detection with strong specificity, high sensitivity, and low purity requirements. Maan et al. [66] reported a real-time reverse transcription polymerase chain reactions (RT-PCR) assay to recognize Seg-2 of the eight epizootic hemorrhagic disease virus (EHDV) serotypes [67,68]. However, such detectors are difficult to implement for point-of-care testing. An immunofluorescence assay (IFA) could also be used for virus detection. Darwish et al. [69] designed a sensitive and specific optical immunosensor for detecting dengue virus markers using a fluorescent signal label with quantifiable signals [70]. Surface-enhanced Raman Scattering (SERS) [71,72] has emerged as a promising tool to guide cancer diagnosis and synergistic therapy. However, SERS requires the use of nanomaterials to enhance the fluorescence signal, and the detection equipment is relatively expensive. Yin et al. [73] reported an ultra-sensitive, highly selective, and SERS-based SARS-CoV-2 simulation sequence (N-cDNA) detection platform. The platform used a magnetic field to control the distance at which superparamagnetic iron oxide nanoparticles (DNA2/DNA3−SPIONs), DNA2-bound N-cDNA, and AuNPs coupled with DNA3 and their complementary sequence (DNA4) are bound, which effectively enhances the fluorescence signal, thereby increasing its sensitivity.

However, this method is too complicated, time-consuming, and expensive, as is the case with electrochemical biosensors [74]. Currently, the SiNW-FET-based biosensor, as an emerging biological detection technology, could be used for protein [54], DNA [75], and microorganism detection [34], providing new solutions for infectious diseases and other health aspects. To date, SiNW-FETs have successfully detected many viruses, including Dengue [4], Influenza A H3N2 [76], H1N1 [77,78], H5N1 [79], and H5N2 [80]; they have also detected two different viruses simultaneously [81].

To improve the detection sensitivity, researchers improved the SiNW-FET biosensor from different aspects. Patolsky et al. [27] functionalized the nanowire array and observed the changes in electrical conductivity before and after binding in order to detect influenza A labeled with fluorescence. The binding of the viruses resulted in a decrease in the electrical conductivity, which returned to baseline when the viruses were released. Meanwhile, their electrical measurements with antibody-functionalized nanowire field effect transistors (NWFET) illustrated that single viruses can be detected in parallel with high detection selectivity. 

Since silanaldehydes are linked to antibodies by reductive amination, it is difficult to reverse the amine link to the original free aldehyde state after antigen detection. Therefore, removing the tightly bound antigen–antibody complex is difficult, resulting in sensors that can only be used once. Chiang et al. [80] proposed a reversible surface modification technology using a disulfide linker, where the receptor molecules were anchored in an MPTMS/SiNW-FET surface with the disulfide bond. Figure 6 shows the experimental setup of the SiNW-FET system, a schematic diagram, and experimental results. The detection limit was 10^−17^ M for H5N2 avian influenza virus (AIV) allergic testing. The sensing device was reusable by fixing the antibody to the SiNW-FET via disulfide bonds and then reducing the disulfide bonds with dithiothreitol (DTT) while removing the antibody–virus complex. This reversible surface functionalization by reducing disulfide bonds is comparable to another technique that modifies the surface of SiNW-FET by reversible binding–dissociation between glutathione (GSH) and glutathione S-transferase (GST)-labeled proteins [17].

The interaction between hemagglutinin (HA) on Influenza virus’ surface and glycans on the host cell was discussed by Hideshima et al. [23]. They were the first to detect H1 and H5 of influenza A viral HA molecules with glycan-immobilized FET biosensor.

In addition to enhancing surface modification, some improvements were introduced during hardware manufacturing. Borgne et al. [82] developed a simple and low-cost bacteria sensor with a self-assembled gas liquid solid method (VLS). It could be used for label-free and ultra-sensitive bacteria detection. Borgne and his colleagues also developed an SiNW-FET-based resistor fabricated by the VLS method and compatible COMS silicon technology. It was found that Escherichia coli bacteria were able to attach precisely to the SiNW networks, which resulted in a drastic decrease in the electrical resistance of the device [83]. Kim et al. [81] improved the sensitivity of biosensors through material selection. One-dimensional SiNW with a nanoscale three-dimension transistor structure was fabricated through isotropic and anisotropic patterning. This sensor could be used for the multiplex detection of AI and HIV viruses.

SiNW-FET, as an emerging detection method, is applied in the field of biological aerosol research. Shen et al. [76] developed a real-time monitoring system for airborne influenza H3N2, integrating electronic-addressing SiNW sensor devices, microfluidics, and bioaerosol-to-hydrosol air sampling techniques. The experimental results showed that the device can rapidly detect influenza A (H3N2) viruses by alternating clean air samples, indoor air samples, and airborne virus samples with a detection limit of 10^4^ viruses/L and specifically differentiate between H3N2 and H1N1 viruses. The optical image chip and microfluidic channel of SiNWs are shown in Figure 7. This monitoring system was tested with an antenna and network. The remote platforms (such as cell phones and computers) observed continuous conductivity changes in seconds, demonstrating the ability to monitor the influenza virus in real time.

The outbreak of the COVID-19 pandemic started in late December 2019 and then spread quickly worldwide, leading to a global public health emergency. Hu et al. [84] proposed an SiNW-FET platform for the detection of IL-6, a disease marker of COVID-19. By using anti-IL-6 aptamers to detect IL-6, SiNW-FET can achieve effective detection concentrations as low as 2.1 pg/mL (100 fM). Moreover, the proposed aptamer-functionalized SiNW-FET detection range is sufficient to measure IL-6 expression levels in COVID-19-infected patients and distinguish between mild and severe disease.

## 5. Challenges and Opportunities

The SiNW-FET biosensor has ultra-sensitive and highly selective properties. It has been widely used in disease diagnosis [85], drug discovery, food monitoring [86], and environmental detection [82]. Table 1 summarizes the sensing performance of silicon nanowire-based FETs for the analysis of biomedical biomarkers. As can be seen from Table 1, the outstanding advantage of SiNW-FETs is their high sensitivity, i.e., ultra-low detection limit. These SiNW-FETs are capable of detecting nucleic acids, proteins, and microorganisms at fM levels. Despite these advantages, the commercialized CMOS-compatible SiNW-FET biosensor is still in its preliminary stage and faces some challenges. The main challenges include the following: (1) how to reduce the Debye shielding effect to improve the sensor sensitivity. The Debye shielding effect occurs when an SiNWs-FET detects charged targets in the sample. When the charge thickness generated by the charged target exceeds the charge thickness formed in the sample solution, the potential field formed by the target will be shielded, thus failing to attract carrier changes in the SiNW channel and reducing the sensitivity of the sensor. (2) Modification and reversibility of the biorecognition layer. Reversible surface modification technology can effectively improve the reusability of SiNW-FET sensors. However, the removal of covalent bonds on the SiNW surface by oxygen plasma or strong oxide can damage SiNWs and reduce the signal of the sensor (the magnitude varies from uA to nA). (3) How to achieve multitarget sensitive detection. Most SiNW-FET sensors can only be used to detect a single target, whereas diseases are often accompanied by multiple biomarkers. Microfluidic technology enables SiNW-FET sensors to detect two or more biomarkers at the same time, which is conducive to the diagnosis of diseases.

### 5.1. Sensor Sensitivity

SiNW-FET biosensors are mainly subject to the interacting electric field imposed by the bound molecular charges to conduct conductance modulation. The sensor sensitivity will reduce due to the Debye–Hückel screening effect. [87,88] It is difficult to trace analytes in high-salt buffer solutions. The whole blood sample [89] within a high-ionic-strength serum should be desalted before a sensor analysis [2,90].

So far, several approaches to overcome the Debye–Hückel screening effect have been proposed, such as the following: (1) replacing antibodies with aptamers [91] as biorecognition functions on silicon nanowires; (2) adding a layer of biomolecule-permeable polymers to the semiconductor material to change dielectric constant [92]; (3) purifying serum and untreated blood by an antigen–antibody binding reaction [93]. The above methods can shorten the distance between the target molecule and the surface of the silicon nanowire or increase the Debye length, which can partially overcome the Debye screening effect. However, this weakens the advantages of sensor rapidity and selectivity. 

It is essential to improve the sensitivity of SiNW-FET biosensors [94] for biomarker detection. One of the challenges scientists face is how to take into account the characteristics of high sensitivity, rapidity, and good selectivity of SiNW-FET biosensors.

### 5.2. Surface Modification

APTES is often used in the surface modification of SiNW-FET biosensors to generate amino groups, linking the corresponding antibody or DNA of the target analytes to the sensor surface. However, low modification efficiency and antibody interactions will affect the detection result. Although cryo-electron microscopy (cryo-EM) technology [95] can distinguish structural information among organic, biological, and nanodevices, it still needs to improve the surface probe modification technology for accurate sensor detection.

In addition, one of the main challenges in the area of biosensor detection is not only the immobilization of the receptors or biological entities of target analytes, but also their complete removal from the surface to promote the reusability of the sensor platform [96]. SiNW-FET could be reusable for different biomolecules because the modified biorecognition layer can be removed by treating the SiNW surface with ultraviolet radiation (UV)/O_2_ plasma. Current studies have focused on the temperature-dependent reversible adhesion of cells on the surface of a stimulatory polymer PNIPAAM-modified sensor. It was shown that when detecting the signal from osteosarcoma SAOS-2 cells, the electrical signal remained constant after each reversible adhesion or dissociation of the cells. Thus, the use of stimulating polymers to modify sensor surfaces can facilitate or hinder the adhesion of biomolecules (e.g., proteins and cells), opening up new possibilities for biosensing applications.

**Table 1 sensors-23-06808-t001:** Silicon nanowire-based FET biosensors for various biomarkers and their analytical performances.

Channel Material	Substrate Material	Analyte	Recognition Element	LOD	Linear Range	Ref
N-type phosphorus-doped SiNW	Si/SiO_2_	DNA	DNA probe	0.1 fM	0.1 fM–10 nM	[26]
P-type boron-doped SiNW	Si/SiO_2_	α-fucosidase	Fuconojirimycin	1.3 pM	1.3 pM–500 pM	[97]
SiNW	SOI wafer	Tnl	CaM	7 nM	10^−8^ M–10^−6^ M	[17]
N-type polycrystalline SiNW	Si/SiO_2_	APOA2	Antibody	6.7 pg/mL	9.5 pg/mL–1.95 ug/mL	[98]
N-type polycrystalline SiNW	Si/SiO_2_	PSA	Antibody	5 fg/mL	5 fg/mL–500 pg/mL	[91]
SiNW	SOI wafer	DNA	DNA probe	0.83 fM	10 fM–10 nM	[99]
P-type boron-doped SiNW	SOI wafer	DNA	Aptamer	1 pM	1 pM–10 nM	[65]
SiNW	SOI wafer	HA	CMP-NANA	1 fM	10^−18^ M–10^−8^ M	[78]
P-type polycrystalline SiNW	Si/SiO_2_	ferritin	Antibody	50 pg/mL	50 pg/mL–500 ng/mL	[100]
SiNW	SOI wafer	cTnl	Antibody	92 pg/mL	92 pg/mL–46 ng/mL	[101]
SiNW	SOI wafer	AFP	Antibody	100 ng/mL	100 ng/mL–500 ng/mL	[102]
P-type boron-doped SiNW	SOI wafer	VEFG	Antibody	5 fM	5 fM–200 fM	[103]

### 5.3. High-Throughput Multiplex Detection

The detection of multiple biomarkers using SiNW-FET sensors has been investigated [81]. For example, the integration of SiNW-FET sensors with Polydimethylsiloxane (PDMS) microfluidic devices can simultaneously detect microRNA and CEA [104]. PDMS-based channels are widely used in SiNW-FET biosensors, which brings the following problems: (1) particles flowing in the channels are difficult to access the nanowire surface; (2) PDMS can adsorb biomolecules, which may degrade the sensitivity of the sensor.

Scientists have also investigated improving microanalysis systems by improving microfabrication and microfluidic device technologies [85,105]. However, high-efficiency, high-sensitivity, high-throughput detection has not yet been achieved [106], which is urgently needed for precision in medical diagnosis and monitoring.

### 5.4. SiNW-FET Device

Silicon nanowires (SiNWs) have attracted much attention in recent years due to their excellent physical and chemical properties. Because SiNW has a high surface-to-volume ratio, Cui et al. [5] produced highly sensitive SiNW sensors to detect biological and chemical molecules. It was found that the design of SiNW-FETs, including SiNW’s size, the threshold voltage, and the manufacturing process, plays a crucial role in the detection performance [20]. The number of SiNWs is positively proportional to the sensitivity, and the size is inversely proportional to the sensitivity. He et al. [107], respectively, designed SiNW-FETs with 1–3 μm spacing between the SiNWs used for detecting the miRNA. The study results found that the SiNW-FET sensor with 1 μm spacing showed superior performance. Mohd et al. [108] proposed that tunnel field-effect transistors (TFETs) had a lower threshold voltage than FETs. Furthermore, the Schottky barrier (SB) in the source/drain was used in the design of TFETs to eliminate the doping effect of the S/D area, thus giving the TFETs better detection performance than the FETs. In addition, the introduction of high-k oxide [109] or charged plasma in TFETs [110] and the introduction of cavities on the source side for sensing biological molecules using dielectric modulation are helpful to enhance the drain current.

The above design method provides a good strategy for improving the detection performance of SiNW-FET sensors. Optimizing the size of the SiNW can enhance the sensitivity of SiNWs to the electric field, and the good detection performance of TFETs provides a solid foundation for improving the performance of SiNW-FET biosensors in the future.

## 6. Summary

An SiNW-FET biosensor with remarkable electronic properties is a promising biosensor for biological analysis. Changes in the internal charge distribution of the device can be sensitively detected by field-effect sensing-based SiNW-FETs. The capture of the target can cause a change in the number of carriers in the device (consumption or accumulation) and finally achieve conductance modulation.

Due to the Debye–Hückel screening effect [88], a highly sensitive strategy should be taken for biomarker detection during cancer [17] and infectious disease [68] testing. In this paper, we discussed the development process of the SiNW-FET biosensor. The detection strategies of the SiNW-FET biosensor are different for nucleic acid molecules, proteins, and microorganisms. When SiNW-FET biosensors detect DNA with PNA probes [34,111], the detection sensitivity is higher than that of DNA probes. This method could be used for virus detection [112], disease diagnosis [113], and drug therapy [114]. In addition, the SiNW-FET biosensor could be used for protein detection through the specific recognition of antigen–antibody/biotin–avidin. The detection sensitivity could be improved with sample pre-treatment (such as ion dilution [29]), chip integration, and multichannel 3D design [55]. Additionally, for microorganism detection, i.e., viruses and bacteria, the SiNW-FET biosensor can be used with a disulfide bond reversible surface modification technique [80] combined with an improved manufacturing process (such as gas liquid solid VLS [115] combined with silicon CMOS technology [116,117] and anisotropic and isotropic etching [20]). The sensitivity of an SiNW-FET biosensor is limited by factors such as serum sample desalination, small quantities of the mixed sample, a complex detection matrix, and probe sensitivity. Future research will focus on the sensitive detection of multiple biomarkers simultaneously for more accurate cancer prevention and clinical sample information.

In summary, the SiNW-FET biosensor is simple, rapid, and sensitive and can be used for real-time DNA, protein, and microorganism detection. An increasing number of portable and low-cost SiNW-FET biosensors are being developed for practical applications, demonstrating their great potential for clinical diagnosis, such as tumor testing, virus determination, and blood testing.

## Figures and Tables

**Figure 1 sensors-23-06808-f001:**
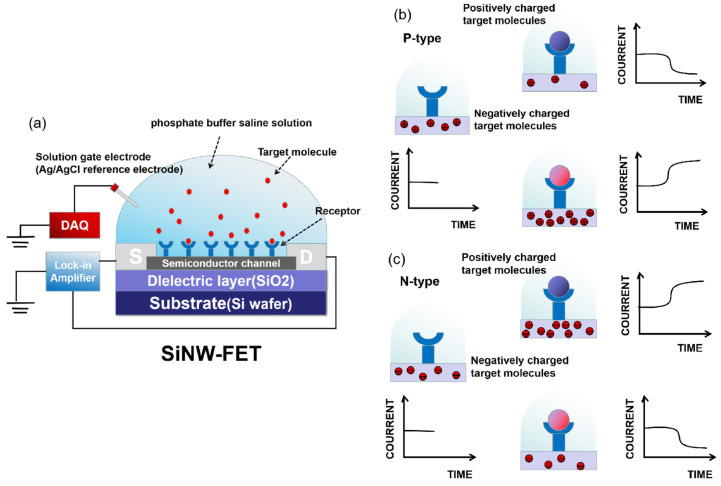
Schematic diagram and working principle of SiNW-FET biosensor. (**a**): The SiNW FET consists of three electrodes: the source electrode, drain electrode, and gate electrode (solution gate). The data acquisition (DAQ) system supplies a gate voltage Vg via inserting a platinum electrode (reference electrode) directly into the solution. The electrical measurements of SiNW-FET are conducted with a lock-in amplifier at V_SD_ = 10 mV, a modulation frequency of 79 Hz, and a time constant of 100 ms. (**b**,**c**): The conductance between the source and drain varies with time for different charged target analytes and different types of semiconductor materials.

**Figure 2 sensors-23-06808-f002:**
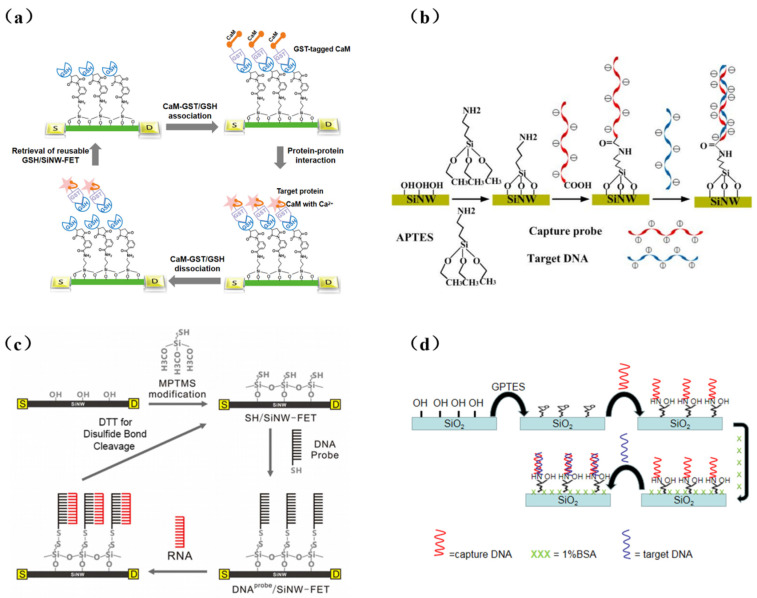
(**a**) The strategy with APTMS to detect proteins [16]. (**b**) Surface modification process of SiNW-FET biosensor for DNA detection with APTES [25]. (**c**) A flow diagram of a reusable DNA probe/MPTMS-modified SiNW-FET device [37]. (**d**) Schematic of the surface modification of nanowires using GPTES [38].

**Figure 3 sensors-23-06808-f003:**
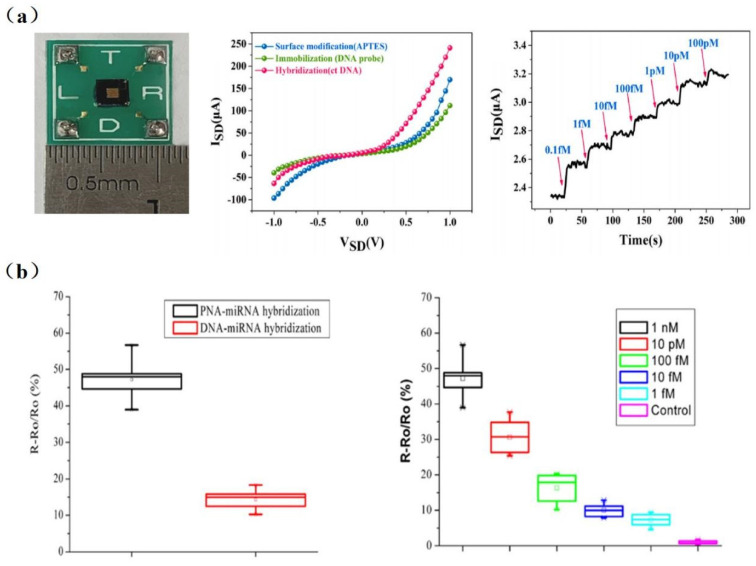
(**a**) Left to right: Photograph of packaged SiNW-array FET devices; I−V characteristics of the same SiNW-array FET sensor at different stage; the real-time response of SiNW-array FET biosensor for different concentrations of ctDNA (from 0.1 fM to 100 pM). Red arrows indicate the points of injections [43]. (**b**) Left to right: The response comparison of DNA-functionalized and PNA-functionalized SiNW biosensors to detect complementary miRNAs; the response of PNA-functionalized SiNW biosensors to different concentrations of complementary miRNAs. Its detection can be as low as 1 fM [32].

**Figure 4 sensors-23-06808-f004:**
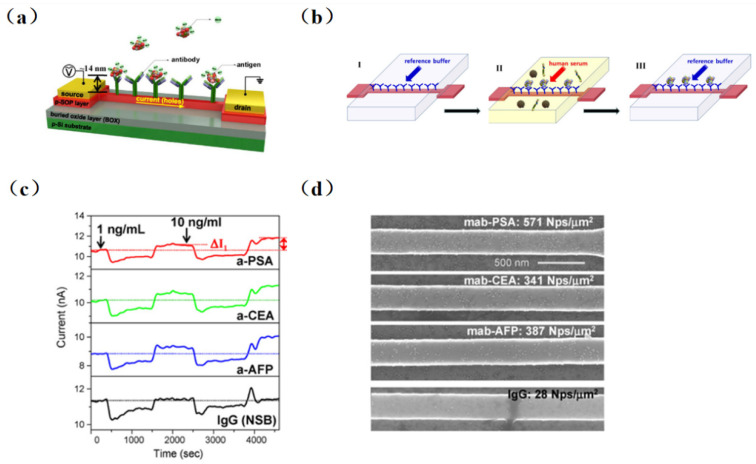
Biosensor schematic diagram and sketch of new sensing method in physiological solution. (**a**) Immobilization of antibodies on the sensing channel between the source and drain electrodes to detect negatively charged antigens. (**b**) Schematic diagram of the novel immunoassay device. In step I, a reference buffer solution (10 μM phosphate, 20 μM NaCl, pH 7.8) is injected to fix the antibody on the surface of the sensor. Step II is for Ag–Ab interaction of target antigen (cancer marker) in the serum containing the immobilized antibody. Step III: the target antigen is bound to the antibody fixed on the FET sensor by buffer solution washing out the human serum, and the electrical measurement can be performed. (**c**) Multiplexed detection of cancer markers: human IgG (black line), PSA (red line), CEA (green line), and AFP (blue line). (**d**) SEM images of four SiNW-FET sensors incubated in immunogolds conjugated with PSA, CEA, and AFP after the multiplexed electrical measurements [53].

**Figure 5 sensors-23-06808-f005:**
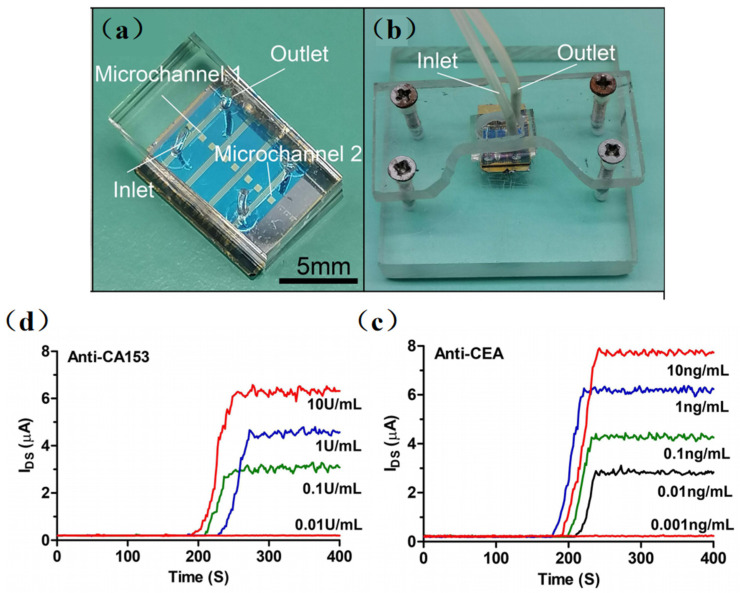
(**a**) Optical image of the PDMS microfluidic device integrated with the SiNW-FET chip. (**b**) Optical image of the SiNW-FET biosensor integrated with acrylic fastening fixture. (**c**) The anti-CA15-3 SiNW-FET biosensor response to different concentrations of CA15-3 in 0.01 × PBS solution. (**d**) The anti-CEA SiNW-FET biosensor response to different concentrations of CEA in 0.01 × PBS solution. [55].

**Figure 6 sensors-23-06808-f006:**
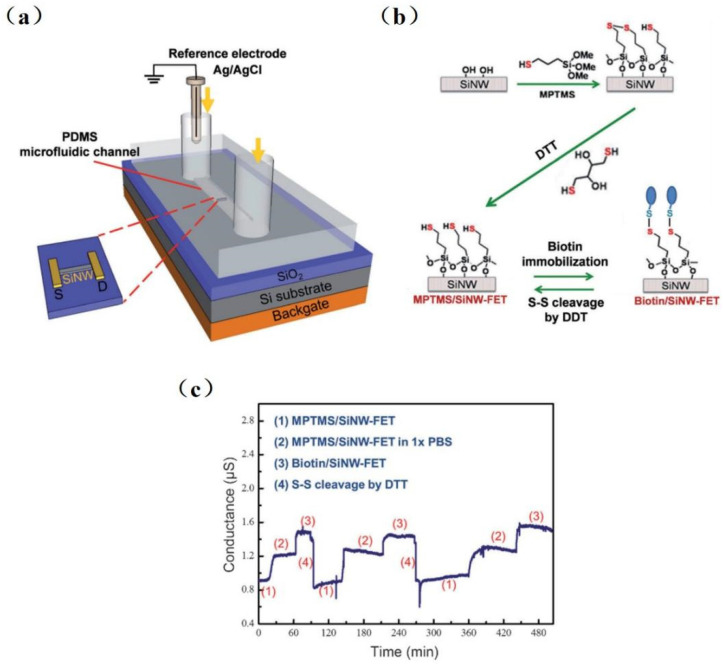
Schematic diagram of the reversible surface functionalization. (**a**) An experimental setup of SiNW-FET system. (**b**) The surface of SiNW-FET was modified with MPTMS and then washed with DTT to form an MPTMS/ SiNW-FET. Biotin-HPDP is fixed on MPTMS/SiNW-FET by disulfide bond, and the disulfide linker is subsequently cracked by DTT, and then the device surface returns to the state of MPTMS/SiNW-FET. (**c**) The reusability of SiNW-FET is proven by three repeatable-period experiments. (1) DTT washes a MPTMS/SiNW-FET, then (2) it is immersed in 1 × PBS and reacts with biotin-HPDP to form (3) biotin/SiNW-FET, and last, the biotin was removed by DTT washing to return [79].

**Figure 7 sensors-23-06808-f007:**
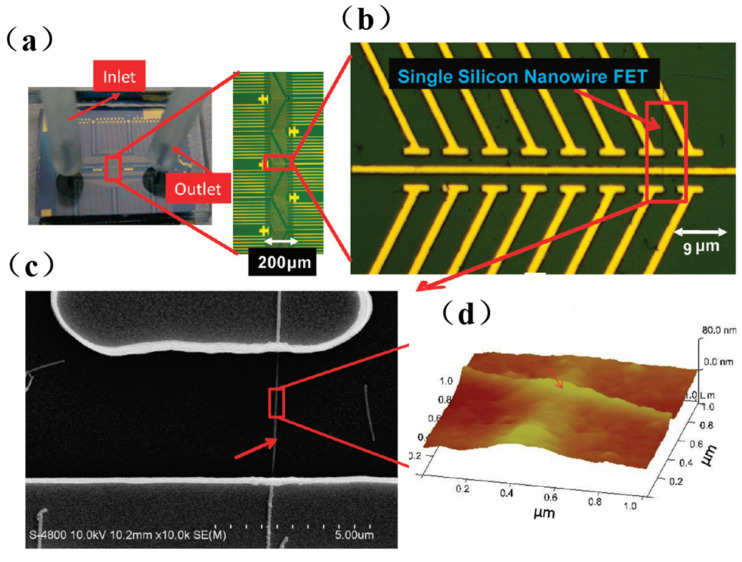
Representative optical images of silicon nanowire (SiNW) chip and microfluidic channel: (**a**) The microfluidic channel of the SiNW sensor array with one inlet and one outlet. (**b**) The enlarged portion (red rectangle (A)) of the SiNW sensor device. (**c**) The SEM image of a single SiNW-FET indicated by arrows in (**b**). (**d**) Atomic force microscopy of antibody-modified silicon nanowires [76].

## Data Availability

Not applicable.

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
