# Peer review of "Application of Silicon Nanowire Field Effect Transistor (SiNW-FET) Biosensor with High Sensitivity"

_sensors, 2023, doi:10.3390/s23156808_

Round 1

Reviewer 1 Report

This topic is interesting for realtime detection of biomolecules. The explanation of working mechanism is clear. The examples are very selected. Overall, this review is reader-friendly. I recommend publish this review after addressing the following issues:

1.       The introduction part is confusing. For line 41 to 42, what does time-consuming refer to and what kind of conformation changes/steric hindrance is causing to what effect?

2.       Apart from chemical or biosensors, other nanoparticle-enhanced techniques such as SERS (e.g., ACS Applied Materials & Interfaces 2022, 14 (3), 4714-4724; Small 2023, 19 (6), 2206762) and fluorescence (e.g., Aggregate 2023, 4 (1), e195; Biosensors and Bioelectronics 2023, 230, 115270), and other mechanisms. The authors should cite these references and discuss their advantages and disadvantages compared to FET technique.

3.       “…SiNW-FETs emerged as the best technology in the field” this claim is highly doubted

4.       Figure 1 can include hole depletion and accumulation processes, as mentioned by the authors

5.       Many figures have low resolution

6.       Line 160-161, explain why UV/O2 plasma can enhance the efficiency of silanization

7.       The author should mention whether DNA extraction is necessary before the assay

8.       The detection limits attaining 0.1 fM by reference 46 and 10 aM by reference 42 are striking. This study should be highlighted in the main figures and a more detailed discussion. Are these numbers determined by 3*background or experimentally?

9.       Please check if the figure reprint permissions are obtained

10.   The authors should discuss whether SiNW-FET are re-usable for different biomolecules

11.   Can SiNW-FET detect target molecules in complex samples, such in blood and urine?

12.   For part of microorganism detection, are these devices detection them by nucleic acid detection of the membrane of the viruses or bacteria?

13.   This review lacks the design and discussion of how detection performance of SiNW-FET can be enhanced.

The English language of this paper needs vigorous revision.

Reviewer 2 Report

The submitted work on a review of SiNW-based FET for biosensing application. The review is detailed and coverup most of the biosensing applications. However, there are a few points missing in the work as per the below-mentioned comments. I would recommend the work given that the authors must clarify a few minor concerns.

Doubts:

  1. The work mentions that SiNWs possess excellent optical, magnetic, and electronic properties, but it would be helpful if the authors provided specific details about these properties and how they contribute to the improved detection sensitivity of biosensors. Elaborating on these aspects would enhance the reader's understanding of the advantages of SiNWs in biosensing applications.

  2. While the work briefly mentions the reversible surface modification methods, it does not provide sufficient information about the techniques used or their significance in the context of SiNW-FET biosensors. Including more details on these methods, their advantages, and any limitations would provide a clearer picture of the current progress in the field.

  3. The work mentions the applications of SiNW-FETs in DNA, protein, and microbial detection, but it does not mention the specific advantages or breakthroughs achieved in these areas. It would be beneficial for the authors to provide examples or case studies illustrating the successful utilization of SiNW-FETs in each of these applications to further support their claims and provide practical insights.

  4. The review discusses the related working principle and technical approaches, but it does not specify which aspects are covered. It would be helpful if the authors provided more information on the specific working principles and technical approaches discussed in the paper to give readers a better understanding of the content covered.

  5. The review provides an extensive discussion of the challenges in the future development of SiNW-FETs. However, it does not provide any indication of what these challenges are or how they impact the field. Including a brief overview of the anticipated challenges and their implications would help readers grasp the significance of future research directions in this area.

  6. Kindly cite the latest work going on biosensor simulations as well which explains the physics behind the sensing such as 10.1109/JSEN.2020.3004050, 10.1149/11101.0249ecst, 10.1109/TNANO.2022.3178845, 10.1016/j.sse.2022.108525, 10.1109/JSEN.2019.2900092, 10.1109/JSEN.2019.2944885, etc. Authors can choose similar works with relevance to the article.

Addressing these doubts would strengthen the manuscript by providing more specific details, clarifying the significance of the discussed topics, and enhancing the overall understanding of SiNW-FETs for biosensing applications.

minor English formatting with correct grammar

Round 2

Reviewer 1 Report

The authors have addressed my issues

N/A